# Individual and contextual predictors of emergency department visits among community-living older adults: a register-based prospective cohort study

Mahwish Naseer ![ORCID] ,[1,2] Kevin J McKee,[1] Anna Ehrenberg,[1] Pär Schön,[2] Lena Dahlberg ![ORCID] [1,2]

¹School of Health and Welfare, Dalarna University, Falun, Sweden
²Aging Research Center, Department of Neurobiology, Care Sciences and Society, Karolinska Institutet, Stockholm, Sweden

**Correspondence to**
Mahwish Naseer; mna@du.se

## ABSTRACT

**Objectives** To examine the extent to which contextual factors explain emergency department (ED) visits and ED revisits, additional to that explained by individual factors.

**Design** A register-based prospective cohort study.

**Setting** Swedish region of Dalarna.

**Participants** Participants were 16 543 community-living adults aged 80 or older who were residents of the Dalarna region of Sweden, excluding older adults who moved out of Dalarna or into residential care during the study period.

**Outcome measures** Dependent variables were initial ED visit, and at least one ED revisit within 30 days of an initial ED visit.

**Results** Approximately 36% of the participants visited the ED during the study period with 18.9% returning to the ED within 30 days. For both initial ED visits and ED revisits, the addition of contextual factors to models containing individual factors significantly improved model fit (p<0.001; p<0.022) and the amount of variance explained in the outcome. In the final models, initial ED visit was significantly associated with older age, number of chronic diseases, receipt of home help, number of primary care visits, proportion of 80+ in the population and shorter distance to the ED; while an ED revisit was significantly associated with greater use of social care, number of hospital admissions and disposition (discharged; admitted to hospital) at initial ED visit.

**Conclusion** Contextual factors explain variance in initial ED visit, additional to that explained by individual factors alone, which indicates inequitable access to ED care. These findings suggest considering local variations in contextual factors in order to improve health-related outcomes among older adults.

## INTRODUCTION

Emergency department (ED) visits are increasing worldwide, resulting in crowding and concern about the appropriateness of such visits.[1] A systematic review on ED visits found that older adults accounted for 12%–21% of all ED visits.[2] In Sweden, adults aged 80 years or older constitute 5% of the total population yet account for 18% of all ED visits.[3] Older adults' complex health needs are associated with a relatively high use of health services but do not necessarily imply a requirement of ED care,[2] which is often poorly adapted to such needs and may carry a risk of negative outcomes.[4 5] While ED providers offer services/treatment for health problems and injuries that require immediate attention, some ED visits are partly explained by an unmet need for primary and social care for older adults[2 6]

The well-established Andersen model of health service use proposes the consideration of contextual factors along with individual factors for a better understanding of healthcare use.[7] Briefly, contextual factors include community characteristics and service provider-related factors that are measured at an aggregated level. Contextual factors, in the same way as individual factors, have been divided into three major components, that is, factors that predispose (eg, proportion of 80+ in the total population), enable (eg, supply of social and healthcare facilities), or approximate individual need for healthcare (eg, disability rate).[7] In this study, we explored whether considering contextual factors in addition to individual factors can improve our understanding of ED visits among older adults.

**Strengths and limitations of this study**

► Contextual factors were included in this study that sought to explain emergency department visits and revisits, this has rarely been done previously.
► This was a register-based cohort study with robust information on the health and social care use of the entire older adult population.
► Home help receipt was measured using both level and quality of home help.
► Administrative differences in how municipalities report data may impact the reliability of municipal-level data.

Individual factors such as age, gender and health problems could potentially explain ED visits. There are mixed findings on the association between old age and ED visits,[8–10] possibly due to inconsistent methodology. There are higher odds of ED revisits among men.[8 11] Primary and social care play an important role in managing chronic diseases and in the proactive treatment of health problems that might require an ED visit.[2] However, research has shown that home help receipt and greater use of primary care are associated with ED revisits and unplanned hospitalisations.[11 12] While it is logical that greater need for care should be associated with greater use of services, service use does not necessarily ensure that care needs are met.[13] Low staff competence and poor continuity of care entails a risk of unmet needs[14] and may lead to avoidable ED visits. In Sweden, the proportion of registered nurses in municipal social care for older adults is lower than in neighbouring Norway (9% vs 31%),[15] while currently a person receiving home help meets an average of 16 different carers during a 14-day period.[3] Research investigating use of healthcare services among older adults has usually included a measure of home help receipt per se, rather than measuring its level (instrumental services, personal care or both) and quality .[11 12 16]

Contextual factors related to healthcare use include community characteristics such as the population age distribution, geographical location and the organisation of health and social care.[7] In Sweden, health and social care is financed, managed and provided by 21 regions and 290 municipalities, with local variations in the management and delivery of care services. For example, without national guidelines on eligibility, municipalities decide on eligibility criteria for home help and residential care services.[17] This challenges the idea that all citizens in a universal welfare state should have equitable access to high quality care services driven by their need rather than, for example, place of residence.[18 19]

Health and social care systems internationally have undergone substantial changes. For example, Sweden has the lowest per capita hospital bed rate in Europe,[20] contributing to shorter lengths of hospital stay and an increased need of primary care, while there has been a 30% reduction in residential care since 2000.[17] A growing proportion of older adults are thus ageing at home and the availability of home help services is key to their living independently.[21] However, home help provision has not compensated for the decrease in residential care.[22] Instead, there has been an increase in informal care, particularly for older adults with lower education, while those with higher education are more likely to buy private services.[18]

It is important to explore contextual factors potentially influencing ED use among older adults. Or and Penneau found that shorter distance to the ED and unavailability of primary care are associated with ED visits.[23] Studies on social care supply for older adults have shown that higher social care expenditures and greater availability of residential care are negatively associated with ED revisits.[21 24]

However, there are few studies on the association between home help coverage and ED visits.[21]

In summary, while both individual and contextual factors have the potential to explain ED care use in older adults, research has primarily focused on individual factors. Moreover, few studies on ED visits and revisits have considered the level and quality of home help receipt.

The aim of this study is to examine the extent to which contextual factors can improve the explanation of initial ED visits and ED revisits by community-living older adults, compared with that provided by individual factors alone.

## METHODS
### Study design, setting and population
A prospective cohort design was used. The study population was community-living adults aged 80 or older who were residents of the Swedish region of Dalarna on 31 December 2014 (N=17 077). Excluding older adults who moved out of Dalarna (N=65) or into residential care during 2015 (N=469), the analytical sample was N=16 543.

There are 15 municipalities in Dalarna whose population densities vary from 1.0 to 90.1 persons/km². Thus, great geographical variation is expected in healthcare and social care access. There are four hospital EDs in Dalarna, all open 24 hours; one is regional, three are local, of which one is only for psychiatric emergencies.

### Patient and public involvement
Patients or the public were not involved in the design or conduct of study.

### Data sources
This study is based on data from three registers: two national (the Longitudinal Integration Database for Health Insurance and Labour Market (LISA) and the Social Services Register) and one regional (the healthcare database of Region Dalarna), all of which cover their respective entire populations. The registers were linked via encrypted personal numbers that are assigned to all individuals living in Sweden. Municipal-level data were accessed from the publicly available national database Kolada (https://www.kolada.se/), which is managed by the Council for the Promotion of Municipal Analyses.[25]

### Dependent variables
There were two dependent variables (DVs): (1) initial ED visit during 2015 and (2) at least one ED revisit within 30 days of an initial ED visit (both coded as no (0); yes (1)).[9 11] Information on visits at the four EDs was obtained from the regional database.

### Independent variables
#### Individual factors
Individual factors were categorised into two domains: (1) demographic and health characteristics, and (2) home help receipt and healthcare use.

### Demographics and health

Information on age in 2014 based on date of birth, gender and municipality of residence was obtained from LISA. Information on registered diagnosis (based on ICD-10) classification) at inpatient and outpatient visits during 2014 was obtained from the regional database and converted using a validated measure of chronic multimorbidity in older adults into number of chronic diseases.[26]

### Home help receipt and healthcare use

Information on home help receipt status at ED visit (or status in December 2014 for those who did not visit the ED during the study period) was obtained from the Social Services Register, which is updated monthly. Home help receipt status was measured as approvals of home help services. Home help receipt was categorised as no home help, only instrumental services (eg, cleaning, shopping), only personal care (eg, hygiene) and both instrumental services and personal care.

Information on number of primary care visits, specialist care visits and hospital admissions 12 months prior to the initial ED visit (or during 2014 for those who did not visit the ED during the study period) and disposition at ED visit (discharged; admitted to hospital) was obtained from the regional database, the latter independent variable (IV) only analysed in relation to the ED revisit DV.

### Contextual factors

Contextual factors covered municipality-level data from 2015 on: the proportion of persons aged 80+ years in the total population; annual social care expenditures per person aged 80+ years; home help quality; median days in residential care for adults aged 65+ resident in long-term care facilities; and distance to the ED. Distance to the ED which was estimated via Google maps and measured in kilometres as the shortest route from the centre of the municipality of residence to the nearest ED.[23] Since the ED in Säter is only for psychiatric care it was excluded in this estimation.

Two indicators were used of home help quality, based on data from an annual national survey distributed to adults aged 65+ years receiving home help and residential care.[27] Each indicator comprises the proportion of individuals within each municipality who provided only positive responses to questions on home help quality, with three questions in each of two domains: (1) response, trust and safety (do home help staff usually respond well to you; do you trust the staff who come to your home; do you feel safe living in your home with the support of home help) and (2) influence and adequate time (do home help staff usually take account of your suggestions and wishes on how help should be provided; can you usually influence the time at which home help staff come to your place; do home help staff usually have adequate time to perform all their tasks at your place).

### Data analysis

SPSS V.27 for Windows was used to conduct all analyses. Descriptive analyses were performed for all individual factors for the total sample and within municipalities, and for contextual factors at the municipality level. ORs with 95% CIs were calculated for the bivariate associations between individual and contextual factors (independent variables, IVs) and ED visits and ED revisits (DVs).

Subsequently, all IVs with a significant bivariate association with the DVs were used to develop multivariable models of the DVs. First, since the data for ED visits and ED revisits were nested within municipalities, the relevance of nested models was tested by generalised linear mixed models.[28] The variance estimate at municipality level was 0.064 for initial ED visit and 0.019 for revisit, providing an intraclass coefficient of 0.019 and 0.005. Thus, only 1.9% of the variance in initial visit and 0.57% in revisit was explained by municipality of residence, that is, considerably below the conventional level indicating a multilevel analysis is required.[29] Therefore, we proceeded with sequential logistic regression.

Model building proceeded in three steps. At the first step, individual demographic and health characteristics were entered in the models. At the second step, individual home help receipt and healthcare use were entered in the models and changes in model fit statistics examined. At the third step, contextual factors were entered in the models to determine if there was significant improvement in the models after entry of all individual factors. The deviance statistic (−2LL), Cox and Snell's $R^2_{CS}$ and Nagelkerke $R^2_N$ test were used for model fitting.[30] The deviance describes the unexplained variance in the model, so the smaller the value of deviance, the better the model fits the data. $R^2_{CS}$ and $R^2_N$ are approximations to the $R^2$ statistic for multiple linear regression which describes the variance in the DV explained by the model. $R^2_{CS}$ and $R^2_N$ are calculated differently and may provide divergent estimates, thus both were included.[30]

Variance inflation factor (VIF) was used to check potential multicollinearity among IVs. VIF values were below 10 for all the IVs and considered acceptable.[31]

## RESULTS
### Descriptive analyses

The mean age of the participants was 85.2 years and 58.5% were women (table 1). The mean number of chronic diseases was 2.1 (range 1.8–2.4 across municipalities). Seventy-four per cent lived at home with no home help, while 5.1% received instrumental services only, 3.9% personal care only and 16.3% received instrumental services and personal care (range for home help receipt 19.1%–37.1%).

The mean number of primary care visits was 9.0 (range 7.5–10.4), while the mean number of specialist care visits was 2.7 (range 1.7–4.1) (table 2). The mean number of hospital admissions was 0.3 (range 0.3–0.4). In the sample, 36.4% had at least one ED visit, of whom 18.9%

**Table 1** Demographic, health and home help use characteristics for total study participants (N=16 543) and stratified by municipality

| Municipality | n | Demographics | | Health | Home help | | | |
|---|---|---|---|---|---|---|---|---|
| | | Age | Female gender | No of chronic diseases | No | Instrumental services | Personal care | Instrumental and personal care |
| | | Mean (SD) | % | Mean (SD) | % | % | % | % |
| Avesta | 1474 | 85.1 (4.1) | 57.1 | 2.4 (2.0) | 80.5 | 3.1 | 2.1 | 14.2 |
| Borlänge | 2599 | 85.1 (4.2) | 57.5 | 2.0 (1.8) | 72.0 | 1.3 | 12.5 | 14.2 |
| Falun | 2925 | 85.2 (4.2) | 58.5 | 2.2 (1.8) | 62.9 | 14.0 | 1.1 | 22.0 |
| Gagnef | 539 | 85.1 (4.0) | 54.2 | 2.1 (1.7) | 79.8 | 2.0 | 4.0 | 14.1 |
| Hedemora | 932 | 85.1 (4.2) | 59.5 | 2.2 (1.9) | 78.8 | 2.3 | 2.0 | 16.8 |
| Leksand | 945 | 85.4 (4.2) | 59.0 | 1.8 (1.7) | 78.7 | 2.7 | 2.7 | 15.9 |
| Ludvika | 1732 | 85.5 (4.3) | 60.9 | 2.1 (1.8) | 75.6 | 6.6 | 2.1 | 15.6 |
| Malung-Sälen | 633 | 84.8 (4.1) | 59.1 | 2.2 (1.9) | 77.3 | 3.4 | 2.3 | 16.9 |
| Mora | 1215 | 85.4 (4.4) | 59.6 | 2.0 (1.9) | 78.2 | 2.2 | 2.8 | 16.7 |
| Orsa | 477 | 85.1 (4.1) | 58.5 | 2.3 (2.0) | 78.8 | 2.5 | 3.1 | 15.5 |
| Rättvik | 905 | 85.3 (4.1) | 60.0 | 1.9 (1.8) | 80.8 | 4.2 | 1.5 | 13.5 |
| Smedjebacken | 617 | 84.8 (4.0) | 55.9 | 2.1 (1.8) | 78.1 | 6.0 | 4.5 | 11.3 |
| Säter | 591 | 85.2 (4.2) | 57.9 | 2.1 (1.8) | 76.3 | 4.0 | 3.0 | 16.6 |
| Vansbro | 440 | 84.9 (4.0) | 58.6 | 2.1 (1.9) | 80.9 | 0.9 | 4.7 | 13.4 |
| Älvdalen | 519 | 85.0 (4.0) | 59.0 | 2.2 (1.9) | 75.7 | 4.2 | 3.6 | 16.4 |
| Total | 16 543 | 85.2 (4.2) | 58.5 | 2.1 (1.8) | 74.6 | 5.1 | 3.9 | 16.3 |

had an ED revisit within 30 days. The mean number of ED visits was 0.7 (range 0.4–0.9).

Contextual factors are presented in table 3. The proportion of adults aged 80+ years varied across municipalities from 5.3% to 9.0%, while social care costs per person varied from Swedish Korona (SEK)207 741 to SEK272,317 during 2015 (approx. US$25 000–US$32 000). Positive home help quality varied between 35.0% and 56.0% for response, trust and safety and between 34.0% and 60.0% for influence and adequate time.

Median days in residential care varied between 407 and 867, while distance to the nearest ED varied from 1.8 km to 78.8 km.

### Initial ED visit

In bivariate analyses, individual factors significantly positively associated with initial ED visit were: age, number of chronic diseases, home help receipt, number of primary care visits, specialist care visits and hospital admissions (table 4). Contextual factors significantly negatively

**Table 2** Healthcare use for participants (N=16 543) and stratified by municipality

| Municipality | N | No of primary care visits Mean (SD) | No of specialist care visits Mean (SD) | No of hospital admissions Mean (SD) | ED visits (Yes) % | No of ED visits Mean (SD) | ED revisit* % |
|---|---|---|---|---|---|---|---|
| Avesta | 1474 | 7.5 (8.0) | 3.0 (8.1) | 0.4 (0.9) | 38.9 | 0.8 (1.5) | 23.5 |
| Borlänge | 2599 | 8.6 (9.3) | 2.6 (8.5) | 0.3 (0.8) | 36.0 | 0.6 (1.1) | 20.0 |
| Falun | 2925 | 10.4 (10.8) | 2.9 (9.3) | 0.3 (0.8) | 41.8 | 0.8 (1.6) | 19.1 |
| Gagnef | 539 | 8.4 (10.3) | 2.2 (6.6) | 0.3 (0.8) | 34.0 | 0.6 (1.2) | 21.3 |
| Hedemora | 932 | 8.1 (8.5) | 2.5 (7.1) | 0.3 (0.8) | 36.8 | 0.7 (1.4) | 22.3 |
| Leksand | 945 | 10.2 (10.9) | 1.9 (3.7) | 0.3 (0.8) | 36.2 | 0.7 (1.3) | 16.2 |
| Ludvika | 1732 | 8.8 (9.8) | 3.0 (9.0) | 0.3 (0.8) | 26.2 | 0.4 (0.9) | 15.6 |
| Malung-Sälen | 633 | 9.2 (9.0) | 2.0 (7.4) | 0.3 (0.7) | 28.1 | 0.4 (0.9) | 13.0 |
| Mora | 1215 | 9.2 (8.6) | 3.3 (9.4) | 0.4 (0.9) | 44.7 | 0.9 (1.6) | 17.8 |
| Orsa | 477 | 8.4 (8.7) | 4.1 (16.8) | 0.4 (0.8) | 44.9 | 0.8 (1.4) | 17.8 |
| Rättvik | 905 | 8.4 (9.0) | 2.3 (6.4) | 0.3 (0.8) | 37.2 | 0.7 (1.2) | 21.1 |
| Smedjebacken | 617 | 8.9 (8.3) | 2.2 (5.5) | 0.3 (0.8) | 26.1 | 0.4 (0.9) | 15.0 |
| Säter | 591 | 9.2 (9.8) | 1.7 (3.8) | 0.3 (0.7) | 33.3 | 0.5 (1.1) | 13.9 |
| Vansbro | 440 | 9.8 (9.6) | 2.2 (7.0) | 0.3 (0.8) | 31.4 | 0.5 (1.1) | 18.4 |
| Älvdalen | 519 | 8.5 (9.2) | 2.6 (12.2) | 0.3 (0.7) | 38.2 | 0.6 (1.1) | 17.6 |
| Total | 16 543 | 9.0 (9.6) | 2.7 (8.5) | 0.3 (0.8) | 36.4 | 0.7 (1.3) | 18.9 |

*Sample for ED revisit includes only those who visited ED (n=5930).
ED, emergency department.;

associated with ED visits were: median days in residential care, proportion of adults 80+ years in the population and greater distance to the ED.

In model 1 with the demographic and health characteristics age and number of chronic conditions included, deviance=20 624.7, $R^2_{CS}$=0.062 and $R^2_N$=0.085. Compared with a constant-only model, model 1 is a significant improvement ($\chi^2(2)$=1065.2, p<0.001). In model 2 with the addition of variables measuring home help and healthcare use, deviance=19 883.9, $R^2_{CS}$=0.103 and $R^2_N$=0.142. Compared with model 1, model 2 is a significant improvement (model

$\chi^2(8)$=1806.1, p<0.001; step $\chi^2(6)$=740.9, p<0.001). In model 3 with the addition of variables measuring contextual factors, deviance=19 701.7, $R^2_{CS}$=0.113 and $R^2_N$=0.155. Compared with model 2, model 3 is a significant improvement (model $\chi^2(11)$=1988.3, p<0.001; step $\chi^2(3)$=182.2, p<0.001. Thus, the multivariable models indicated a better fit at each step.

In the final multivariable model (model 3), individual-level factors with significant higher ORs for ED visits were: age, number of chronic diseases, home help receipt, and number of primary care visits. Contextual factors

**Table 3**  Contextual factors at municipality level

| Municipality | Proportion of 80+ in the population | Annual social care cost per person 80+ | Positive home help quality | | Median number of days in residential care | Distance to the nearest ED |
| | | | Response, trust and safety | Influence and adequate time | | |
| | % | SEK | % | % | | Km |
| Avesta | 6.7 | 207 741 | 41.0 | 44.0 | 578 | 2.3 |
| Borlänge | 5.3 | 229 601 | 37.0 | 41.0 | 585 | 21.3 |
| Falun | 5.4 | 244 487 | 48.0 | 60.0 | 573 | 1.8 |
| Gagnef | 5.7 | 255 825 | 44.0 | 57.0 | 610 | 40.2 |
| Hedemora | 6.3 | 218 694 | 40.0 | 53.0 | 680 | 20.9 |
| Leksand | 6.7 | 235 201 | 50.0 | 51.0 | 561 | 49.3 |
| Ludvika | 7.0 | 224 179 | 42.0 | 46.0 | 715 | 64.4 |
| Malung-Sälen | 7.0 | 223 591 | 51.0 | 55.0 | 867 | 73.0 |
| Mora | 6.4 | 250 077 | 40.0 | 46.0 | 407 | 2.00 |
| Orsa | 7.3 | 221 582 | 37.0 | 34.0 | 740 | 12.1 |
| Rättvik | 9.0 | 209 722 | 45.0 | 49.0 | 597 | 35.6 |
| Smedjebacken | 6.1 | 215 149 | 39.0 | 40.0 | 647 | 60.8 |
| Säter | 5.8 | 229 107 | 56.0 | 53.0 | 506 | 43.2 |
| Vansbro | 7.0 | 219 152 | 35.0 | 39.0 | 828 | 78.8 |
| Älvdalen | 7.5 | 272 317 | 52.0 | 49.0 | 711 | 41.7 |

ED, emergency department; SEK, Swedish Korona.

significantly associated with ED visit were: higher proportion of adults 80+years in the total population and shorter distance to the ED.

### ED revisit

In bivariate analyses, number of chronic diseases, receipt of both instrumental and personal care from home help services, number of primary care visits and hospital admissions prior to initial ED visit were significantly positively associated with ED revisit, while being admitted to hospital at initial ED visit was significantly negatively associated with an ED revisit (table 5). Of the contextual factors, distance to the ED was significantly negatively associated with an ED revisit.

In model 1 with the health characteristic number of chronic conditions included, deviance=5718.9, $R^2_{CS}$=0.004 and $R^2_N$=0.006. Compared with a constant-only model, model 1 is a significant improvement ($\chi^2(1)$=22.4, p<0.001). In model 2 with the addition of variables measuring home help and healthcare use, deviance=5650.0, $R^2_{CS}$=0.015 and $R^2_N$=0.025. Compared with model 1, model 2 is a significant improvement (model $\chi^2(7)$=91.3, p<0.001; step $\chi^2(6)$=68.9, p<0.001). In model 3 with the addition of the contextual factor distance to the nearest ED, deviance=5644.8, $R^2_{CS}$=0.016 and $R^2_N$=0.026. Compared with model 2, model 3 is a significant improvement (model $\chi^2(8)$=96, p<0.001; step $\chi^2(1)$=5.26, p<0.022). Thus, the multivariable models indicated a better fit at each step.

In the final multivariable model (model 3), receipt of both instrumental and personal care from home help

services, number of hospital admissions, and disposition at initial ED visit were significantly associated with ED revisits. Although the addition of distance to nearest ED in model 3 resulted in a significant improvement in model fit compared with model 2, as the CIs of the OR for the variable contains 1 we conclude that the association between distance to ED and ED revisit was not significant in the model.

## DISCUSSION
### Main findings

Approximately 36% of the participants visited the ED, of whom 18.9% had an ED revisit within 30 days. Research has mainly focused on individual predictors of ED care. Our study indicates that ED care can be further explained by contextual factors. Contextual factors associated with higher odds of initial ED visit indicate inequitable access to ED care. Receipt of both instrumental services and personal care is an important predictor of ED visits and revisits.

### Comparison with other studies

Our findings on the associations between age, number of chronic diseases and initial ED visits are in line with Andersen's model, which describes equitable access to ED care explained by demographic factors (eg, age) and need (eg, chronic diseases).[7] Health decline with age and multimorbidity relates to the increasing need for healthcare. Grunier et al argued that social and primary care can play a role in the management of need and can contribute

**Table 4** Models for binary logistic regression of factors associated with initial ED visit (N = 16 543)

| | ED visit | | | |
| --- | --- | --- | --- | --- |
| | Bivariate | Model 1 | Model 2 | Model 3 |
| | OR (95% CI) | OR (95% CI) | OR (95% CI) | OR (95% CI) |
| **Individual factors** | | | | |
| **Demographics and health** | | | | |
| Age | 1.08 (1.07 to 1.09)* | 1.09 (1.08 to 1.09)* | 1.06 (1.05 to 1.06)* | 1.06 (1.05 to 1.07)* |
| Gender | | | | |
| Female | Ref | | | |
| Male | 1.04 (0.97 to 1.11) | | | |
| No of chronic diseases | 1.22 (1.20 to 1.25)* | 1.24 (1.21 to 1.26)* | 1.12 (1.10 to 1.15)* | 1.12 (1.10 to 1.15)* |
| **Home help and healthcare use** | | | | |
| Home help | | | | |
| No | Ref | | Ref | Ref |
| Instrumental services | 1.74 (1.51 to 2.01)* | | 1.31 (1.13 to 1.52)* | 1.26 (1.08 to 1.46)* |
| Personal care | 2.00 (1.71 to 2.34)* | | 1.49 (1.26 to 1.76)* | 1.55 (1.31 to 1.83)* |
| Instrumental and personal care | 3.38 (3.10 to 3.68)* | | 2.49 (2.27 to 2.74)* | 2.48 (2.26 to 2.73)* |
| No of primary care visits | 1.05 (1.04 to 1.05)* | | 1.03 (1.03 to 1.04)* | 1.03 (1.03 to 1.04)* |
| No of specialist care visits | 1.02 (1.01 to 1.02)* | | 1.00 (1.00 to 1.01) | 1.00 (1.00 to 1.01) |
| No of hospital admissions | 1.37 (1.32 to 1.42)* | | 1.01 (0.96 to 1.05) | 1.00 (0.96 to 1.05) |
| **Contextual factors** | | | | |
| Annual social care cost per person 80+ | 1.00 (1.00 to 1.00) | | | |
| Response, trust and safety | 1.00 (0.99 to 1.00) | | | |
| Influence and adequate time | 1.00 (1.00 to 1.01) | | | |
| Median days in residential care | 0.99 (0.99 to 0.99)* | | | 1.00 (0.99 to 1.00) |
| Proportion of 80+ in the population | 0.94 (0.91 to 0.98)* | | | 1.09 (1.05 to 1.13)* |
| Distance to the nearest ED | 0.99 (0.99 to 0.99)* | | | 0.99 (0.98 to 0.99)* |
| Deviance (−2LL) | | 20 624.749 | 19 883.865 | 19 701.645 |
| Cox and Snell $R^2_{CS}$ | | 0.062 | 0.103 | 0.113 |
| Nagelkerke $R^2_N$ | | 0.085 | 0.142 | 0.155 |

Note: The negative bivariate association for proportion of 80+ in the population with initial ED visit becomes a positive association and increases in magnitude in the multivariable model, indicating negative suppression. Through trial variable removal from the model,[30] it was determined that distance to the ED and median days in residential care both acted as suppressor variables for proportion 80+ in the population.
*Significant on values estimated from 95% CI.
ED, emergency department.

positively to reducing ED visits. Our study shows that the addition of home help and healthcare variables increased the variance explained in DVs by demographics and health characteristics and improved the overall model fit. Positive associations between initial ED visit and both home help receipt and primary care visits suggest that the provided services do not meet the recipient's care needs,[2] that is, indicate a poorly functioning health and social care system. However, extensive home help provision is also an indication of poor health/functional status with an increased risk for ED visits.

The needs of older adults are often complex, and inadequate care co-ordination between hospitals, primary and municipal social care for older adults increases the risk of negative health outcomes. In Sweden, as in other high-income countries, the fragmentation of health and social care systems has become more visible during the COVID-19 pandemic, and there is a need to establish

channels that ensure continuous and rapid coordination of services from different care providers and levels of organisations.[32 33] Research has shown that integrated care, that is, co-ordination between different care providers, can reduce ED visits, hospital admissions, and the length of hospital stay.[34 35]

The ED visit models explain more variance in the outcome than the revisit model and more IVs are significant in the model, including contextual factors. This may be partly explained by the revisit models having less variation in its DV, and perhaps also due to ED revisits more than initial ED visit being related to extraneous variables that were not available in the registers accessed and thus not included in the models. The odds of revisits were higher in those who were not admitted to inpatient hospital care at initial visit and among those with greater use of home help services and higher number of hospital admissions. With shorter lengths of hospital

**Table 5** Models for logistic regression of factors associated with ED revisit within 30 days after initial ED visit (N=5930)

| | ED revisit | | | |
| --- | --- | --- | --- | --- |
| | Bivariate | Model 1 | Model 2 | Model 3 |
| | OR (95% CI) | OR (95% CI) | OR (95% CI) | OR (95% CI) |
| **Individual factors** | | | | |
| **Demographics and health** | | | | |
| Age | 0.99 (0.98 to 1.01) | | | |
| Gender | | | | |
| Female | Ref. | | | |
| Male | 1.12 (0.98 to 1.27) | | | |
| No of chronic diseases | 1.07 (1.04 to 1.10)* | 1.07 (1.04 to 1.10)* | 1.02 (0.98 to 1.06) | 1.02 (0.98 to 1.06) |
| **Home help and healthcare use** | | | | |
| Home help | | | | |
| No | Ref | | Ref | Ref |
| Instrumental services | 0.99 (0.75 to 1.32) | | 0.99 (0.74 to 1.32) | 0.97 (0.73 to 1.29) |
| Personal care | 1.13 (0.84 to 1.53) | | 1.13 (0.84 to 1.53) | 1.15 (0.85 to 1.55) |
| Instrumental and personal care | 1.33 (1.15 to 1.55)* | | 1.31 (1.13 to 1.53)* | 1.30 (1.12 to 1.51)* |
| No of primary care visits | 1.01 (1.01 to 1.02)* | | 1.00 (1.00 to 1.01) | 1.00 (1.00 to 1.01) |
| No of specialist care visits | 1.00 (0.99 to 1.00) | | | |
| No of hospital admissions | 1.19 (1.13 to 1.26)* | | 1.15 (1.08 to 1.23)* | 1.15 (1.08 to 1.23)* |
| Disposition at index ED visit | | | | |
| Discharged | Ref | | Ref | Ref |
| Hospital admission | 0.73 (0.65 to 0.83)* | | 0.69 (0.61 to 0.79)* | 0.71 (0.62 to 0.81)* |
| **Contextual factors** | | | | |
| Annual social care cost per person 80+ | 1.00 (1.00 to 1.00) | | | |
| Response, trust and safety | 0.98 (0.97 to 1.00) | | | |
| Influence and adequate time | 0.99 (0.99 to 1.00) | | | |
| Median days in residential care | 1.00 (0.99 to 1.00) | | | |
| Propotion of 80+ in the population | 0.99 (0.92 to 1.06) | | | |
| Distance to the nearest ED | 0.99 (0.99 to 0.99)* | | | 0.99 (0.99 to 1.00) |
| Deviance (–2LL) | | 5718.922 | 5650.039 | 5644.775 |
| Cox and Snell $R^2_{CS}$ | | 0.004 | 0.015 | 0.016 |
| NagelkerkeR$^2_N$ | | 0.006 | 0.025 | 0.026 |

*Significant on values estimated from 95% CI.
ED, emergency department.

stay,[20] home help providers have to care for older adults with more complex care needs.[3] A recent report showed that patients who were sent home after initial ED visit had a declining health status at ED revisit within 30 days, that is, that their needs for aftercare were not met.[36]

Our results show that contextual factors (living in municipalities with higher proportions of adults aged 80+ in the population and a shorter distance to the ED) increased the odds of an initial ED visit at the individual level. Critical events that require ED care are common in the 80+ group due to multiple health problems, complex needs and rapid changes in health status, which put pressure on the local healthcare system and this may explain the association between a higher proportion of 80+ in the population and ED visits. The association between distance to the ED and ED visits raises concern on the appropriateness of such visits. Access to ED care should be driven by need rather than place of residence, and the variance in initial ED visit explained by contextual factors challenges the idea of universalism and equitable access to healthcare. This suggests that considering the healthcare needs of the local population and geographical variation in healthcare planning may improve the accessibility of primary and social care for older adults.[23]

## Implications

Those individuals who receive both instrumental services and personal care constitute a vulnerable group that should be considered in interventions for reducing ED visits, discharge planning, and aftercare. Considering local variations in healthcare needs and access to health

and social care is important for improving health-related outcomes among older adults.

## Strengths and limitations

A strength of this study was that it was based on register data. Registers provide high-quality information on health and social care use covering the entire population. This study also included contextual factors in seeking to explain ED visits, which have been rarely considered previously, as well as levels and quality of home help receipt as factors in ED visits rather than simply use/non-use.

Home help receipt was categorised as an individual factor, while home help quality was categorised as a contextual variable as the individual-level data were only available for analysis as aggregated data. Thus, some variables that might explain ED visits or revisits can be optionally conceptualised and operationalised at individual or contextual levels. An association between variables can differ in strength and direction when analysed at individual vs group level, so it is important that the use of aggregated data as a contextual factor can be defended. The reliability of the data on home help quality will have been influenced by a survey response rate that varied between 48% and 66% by municipality and only 46% of the data provided by care recipients with the rest provided by relatives or care providers.

The measure of the number of chronic diseases used in this study included healthcare use in 2014 only. Patients who were admitted to the hospital compared with those discharged after the initial ED visit did not have an equal chance to be included within the 30 days criterion for an ED revisit. This problem is minor, though, since the median length of stay in hospital was 4 days. Administrative differences in how municipalities report data to the Kolada database may impact data reliability. Distance to the nearest ED was estimated from the city centre rather than place of residence. Finally, the amount of variance our models explained in their respective outcomes was not substantial, this was particularly the case for ED revisits. This is perhaps not surprising given the many factors that might potentially influence ED use and ED revisits, and indicates that future research should seek to incorporate a broader range of variables than were available for this study to advance the understanding of ED visits and revisits.

## CONCLUSION

The aim of this study was to consider whether contextual factors improve our understanding of ED visits. Results suggest that contextual factors explain additional significant variance in initial ED visit compared with that explained by individual factors alone, which indicates inequitable access to ED care.

In individual factors, age and health characteristics explain variance in initial visit while receipt of home help and healthcare use explained variance both in initial visit and revisit. ED revisit was also explained by discharge

from ED and can be interpreted as a result of inadequate care and discharge planning at the initial visit. Greater use of home help and primary care does not ensure that needs are met, and further research on the impact of unmet care needs in older adults on ED visits can clarify how the use of different health and social care services are associated. Local variations in home help receipt and utilisation of ED care challenge the notion of a universal welfare state, and require consideration in order to improve health-related outcomes among older adults.

**Contributors** Planning and study design (MN, KM, AE, PS and LD), conducted data analysis (MN), advice regarding data analysis (KM) interpreting analyses and results (MN, KM, AE, PS and LD), drafting of manuscript (MN), revising and finalising the manuscript (MN, KM, AE, PS and LD), final approval of the manuscript (MN, KM, AE, PS and LD). MN is acting as guarantor of this study.

**Funding** The study was supported by Swedish Research Council for Health, Working Life and Welfare (grant no. 2015-00440) and Region Dalarna.

**Disclaimer** The funding organisation had no role in the development of study design, analysis, and/or in manuscript writing.

**Competing interests** None declared.

**Patient and public involvement** Patients and/or the public were not involved in the design, or conduct, or reporting, or dissemination plans of this research.

**Patient consent for publication** Not applicable.

**Ethics approval** Ethical approval for the study was awarded by the Regional Ethical Review Board in Stockholm (reg.no. 2016/299-31).

**Provenance and peer review** Not commissioned; externally peer reviewed.

**Data availability statement** Data are available in a public, open access repository. Data may be obtained from a third party and are not publicly available. Individual-level data are obtained from registers (anonymised) and are not publicly available. Municipal level data are available publicly at kolada.se.(dataset) 25. Council for the Promotion of Municipal Analyses. Data from: Kolada; Den öppna och kostnadsfria databasen för kommuner och regioner (The free database for municipalities and regions) 2015. Kolada, 1 December 2021. https://www.kolada.se/

**ORCID iDs**
Mahwish Naseer http://orcid.org/0000-0001-7231-826X
Lena Dahlberg http://orcid.org/0000-0002-7685-3216

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
