## [Reviewer comments · BMJ Open]

ARTICLE DETAILS

TITLE (PROVISIONAL)	Individual and contextual predictors of emergency department visits among community-living older adults: A register-based prospective cohort study
AUTHORS	Naseer, Mahwish; McKee, Kevin; Ehrenberg, Anna; Schön, Pär; Dahlberg, Lena

VERSION 1 – REVIEW

REVIEWER	Bury, Gerard University College Dublin, General Practice
REVIEW RETURNED	07-Sep-2021

GENERAL COMMENTS	An interesting contribution to the existing literature on drivers for health service utilisation, in this case ED visits by older persons. The design
---

REVIEWER	Schmiedhofer, Martina Charite Universitätsmedizin Berlin, Arbeitsbereich Notfall- und Akutmedizin
REVIEW RETURNED	13-Sep-2021

GENERAL COMMENTS	Overall, the authors address an important research topic: ED visits as an indicator of the quality of ambulant health care among the population in an age of 80+. To this end, they employ a quantitative model which explains ED visits by individual and, as a novel contribution from the perspective of the authors, contextual factors. The database refers to a region in Sweden which comprises municipalities with a high variance of demographic characteristics and other factors relevant for the model. The findings indicate that considering contextual factors improves the explanatory power of the model. While the statistical analysis is carried out in a consistent way, some questions regarding the research setting and the interpretation of the results remain open and need to be addressed in further detail: First, while it is obvious that individual factors such as higher age and the number of chronic diseases are highly correlated with ED visits, the latter factor can also be an indication for poor quality of outpatient care. The direct and indirect mechanisms by which those factors might affect ED visits should be discussed therefore in detail when the results are presented. Second, albeit the paper focuses on the relevance of context factors, only two of the six factors considered by the original model appear to be significant in the final model: distance of the place of residence to the ED and the proportion of 80+ residents in the
--

	population. That distance between home and nearest ED increases the odds of ED visit is an obvious and plausible result since a smaller distance delivers a more convenient access. The interpretation of this findings seems to be however a bit vague and overstretched: What consequences follow from the general statement that access to care should be driven by need rather than by place of residence? Shall we supply more or less EDs? What can we learn from the finding if distance is always a factor affecting the utilization of EDs even if distance is small? That the proportion of 80+ residents is a predictor for a higher chance to ED visits is an interesting hint. One might suspect that a higher share of elderly in a municipality raises health care costs and, hence, reduces per capita expenses on health care. However, the annual budget costs for social care per person do not appear significant in the statistical analysis. The reader wonders therefore why the proportion of the 80+ residents in the population affects the odds of ED visits. The authors however do not offer an interpretation for their finding. To strengthen the results, some recommendations are made: Including Frequent User of ED (FUED) as indicator for a poor quality of outpatient care FUED (min. 5 visits per year) account for a disproportionately high number of all ED visits and are more likely to be associated with the lack of sufficient outpatient medical treatment, while one revisit might be random. Therefore, the proportion of FUED could enhance the interpretation of the results and should be also covered by the analysis as an additional outcome variable. Stratifying the age group of the 80+ population into further age brackets, e.g. age 80-84, age 85-89, age 90+. Note that the health situation might differ largely across these groups and, hence, the need for health care and ED visits. Considering further age brackets by the analysis could therefore deliver also interesting insights on the context factors, e.g. variables which appear insignificant for the aggregate 80+ population, might appear significant in subgroups. As an example, a community with a higher proportion of elderly residents might include more high-aged people who by nature suffer from poorer health which in turn might lead to a higher frequency of critical conditions.
--	--

REVIEWER	McBride, David University of Otago, Department of Preventive and Social Medicine
REVIEW RETURNED	16-Sep-2021

GENERAL COMMENTS	Thank you for this interesting paper. I did have a little difficulty with 'contextual factors', lying as they do at the heart of your theory. The Andersen paper is the final phase in developing models for studying community health, and the model (using aggregate data) "divides the major components of contextual characteristics in the same way as individual characteristics have traditionally been divided—those that predispose (eg, community, age structure), enable (eg, supply of medical personnel and facilities), or suggest need for individual use of health services (eg, mortality, morbidity and disability rates). It would help the reader if the introduction
--

	explained this, and if you structure the introduction as you have done in the summary (the second paragraph of page 6). This would foreshorten and clarify this important section. It also seems that primary and social care are central to the model. In the methods, you use the Cox and Snell and the Nagelkerke R2 models to assess goodness of fit and improvement in the model, but how much of the variance do you actually explain? In the discussion, it would help to structure it as main findings, strengths and limitations (as you have) comparison with other studies and then meanings and or implications for practice. What are these? It seem to this reviewer that the main finding was that instrumental services and personal services are associated with visits to the ED, but can you really say that this suggests unmet needs? I think not, and you can't really say that it indicates a poorly functioning health service. In strengths and limitations, you should mention the low proportion of the variance that you actually explained. Thank you again.
--	---

REVIEWER	Shade, Kate Samuel Merritt University
REVIEW RETURNED	21-Sep-2021

GENERAL COMMENTS	Thank you for conducting this interesting research. Below are my questions and suggestions with your statements in double- and my suggested revisions in single-quotation marks. Abstract: Explain ED before use and/or use acronym only in text. Revise Participants: 'Participants were 16,543 community-living adults aged 80 or older who were residents of the Dalarna region of Sweden, excluding older adults who moved out of Dalarna or into residential care during the study period.' Use the term participants, rather than sample, throughout. Include exposure/predictor variables as well as outcome variables in the abstract. Wasn't the main outcome of interest one or more ED revisits within 30 days of an initial ED visit? Results: 'Approximately 36% of participants visited the ED during the study period with 19% returning to the ED within 30 days.' Do not include p-values in the abstract; rather, present the effect size with 95% CI. 'In final models, an ED visit was associated with...while a return ED visit was...' Explain in the variables section of the abstract what you mean when you refer to "proportion of 80+ in the population" and "disposition at initial ED visit." Conclusion: 'Contextual factors in addition to individual factors improved prediction of return ED visits among older adults. These findings may indicate inequitable access to ED care and suggest the need for local support to improve health-related outcomes among older adults.' Strengths and Limitations: 'Contextual factors that explain ED use were included as predictors of returning to the ED within 30 days; this was a register-based cohort study, with robust information about the health status and health and social care use for the entire older adult population; receipt of home help was measured using both type and quality of home help care as predictors; administrative differences in how municipalities report data may impact data reliability.' Keywords are not represented in the abstract. Community-living older adults? Contextual predictors of ED use? 30-day ED revisits? Page 4: For the first reference, consider citing a more recent databased-paper rather than a 2012 review article about ED crowding (but not the appropriateness of use). I recommend
--

	'Researchers conducting a systematic review found...' Similar to your first citation, the second is not based on recent data; I suggest this recent systematic review https://bmcgeriatr.biomedcentral.com/articles/10.1186/s12877-019-1197-9 Remove the statement "which is often poorly adapted to such needs and may carry a risk of negative outcomes" if the meaning is explained in the sentences following. Otherwise, explain further the risk for negative outcomes. Note: ED providers offer services/treatments, not EDs—avoid other anthropomorphism e.g. 'We explored' rather than "this study explores." Based on the outcome of interest identified in the abstract, I expected a purpose statement such as 'We explored whether contextual in addition to individual factors explained a return visits to the ED within 30 days for adults ≥80 years of age.' I recommend removing the sentence about your study aim since it is described at the end of the introduction. Please briefly describe "the well-established Andersen model" for those not familiar with the model. You may want to add a statement such as 'we used the Anderson model as the conceptual framework for this research.' Rather than "different care services," focus on your outcome of interest, 'ED use.' I do not think examples are necessary for the next sentence. Again, no need to discuss gender differences in primary care use, unless you link it to ED visits/revisits. Say more about the predictors of ED revisits, as described by de Gelder et al. (2018). Their findings are most relevant to your research. Though the data collected by Hass et al. (2015) are old, their study is also worthy of mention in this section. Their outcome of interest was similar—the number of ED admissions in a year, and their primary independent variable of interest, unmet ADL needs, it seems to me, could be defined as lack of receipt of home help. Perhaps 'logical' or 'commonsense' instead of "appropriate" when discussing the link between the need for (home help) care and (ED and hospital) services? Instead of "Low staff competence and (lack of) continuity (of home help services)," given the information you offer next, I recommend 'A limited number of home help providers and poor continuity of care...' You may want to cite Bravell et al. (2021). Page 5: Para #1: You offer helpful, local information about available home health nurses in Sweden vs. Norway. However, the reference (based only on title) does not seem applicable ("International COVID-19 care in residential care.") Is the source not your reference #3? Do the terms "municipal social care" and "home help" mean the same thing? If they do, I would use home care or home help consistently. Readers may not know what a fortnight is. Instead of "However, studies on health care use have focused on home help receipt per se, rather than its level..." I recommend: 'Most researchers investigating the use of health care services among older adults measure home help receipt per se, rather than type and quality of home help.' You cite your previous work for this statement so perhaps 'We and other researchers investigating...' instead. Para #2: Add 'services' to the end of the first sentence. Instead of "...municipalities decide on eligibility criteria for services provided via home help and residential care," I recommend '...municipalities decide on eligibility criteria for home help and residential care services.' Revise as well '...driven by their need rather than their place of residence.' Based only on a review of the abstract, it does not appear that reference #19 supports your claim. Is there any
--	--

	more recent research about the variability of services for older adults in Sweden? Para #3: Remove the reference to "international changes" in the first sentence since your focus is on Europe. Revise the second sentence so it is two sentences or phrases joined with a semi-colon—the first about hospital use and the second about residential care use. Revise "...home help provision has not fully compensated for the decrease in residential care." I recommend 'Use of home help services has not increased relative to the decreased use of residential care among older adults.' Reference #18, though published in 2012, is based on data that are >15 years old. Is there no more recent reference? Para #4: I suggest the topic sentence should be 'It is important to explore contextual factors potentially influencing ED use among older adults.' Refer to prior study findings in past tense e.g. 'Researchers in France found...was associated with...' Use a similar format for other researchers' findings e.g., 'Researchers in England found limited social care services predicted increased ED revisits.' Page 6: See my prior comment about your purpose statement. I thought the outcome of interest was 30-day ED revisits? Perhaps add a secondary aim about identifying predictors of ED revisits. Para #3: Revise "...resident in the Swedish..." to '...who were residents of the Swedish...' Specify that there were no exclusion criteria other than moving out of the area or into residential care. Specify how and how often participant follow-up data were collected. Include end date for the study period. Para #4: Instead of "All hospital EDs are 24-hour operational, one regional (Falun) and three local (Mora, Avesta, and Säter; Säter is for psychiatric care patients only)" I suggest 'There are 4 hospital EDs in Dalarna, all open 24-hours; one is regional, three are local, and one is only for psychiatric emergencies.' If these services are not long-standing or have changed, I recommend 'During the study period, there were 4...' I do not understand the need for section 2.2. Use consistent upper- or lower-case titles for all headings/subheadings. Page 7: I would discuss variables of interest before describing data sources. Then identify how each variable, with the level of measurement already defined, was extracted from the data source. Are there 3 or 4 registers? Is the Kolada database the same as the Dalarna regional database? Address any bias in terms of data collection or measurement of variables of interest e.g. you mention in the abstract that municipal data may vary across regions. Explain further here. Re DVs: Specify whether ED visits included visits to any ED e.g., psychiatric ED. Since you did not measure the distance to psychiatric ED, I assume you did not include psychiatric ED visits. Explain. Re IVs: Include measurement of each demographic variable e.g., age in 2014 based on date of birth. Revise 'We gathered information about the health of participants by collecting ICD 10 diagnoses for all inpatient and outpatient visits in 2014 from the regional database. We measured the health of participants as the number of ICD codes for chronic diseases.' Note this limitation (that data were based only on 2014 healthcare utilization) elsewhere. Page 8: Specify how often data were collected for receipt of home help during 2014 or 2015. Were data collected about # of home help visits? Type of home help provider? Over what period of time? Until now, I assumed home help included nurse home visits. If this
--	---

	is not the case, it would be helpful to define home help as excluding nurse home visits earlier in the paper. Para #2: I don't understand the statement "...the latter IV only analysed in relation to the ED revisit DV." Data about disposition at initial ED visit was collected only for participants who had a 30-day ED visit? Para #3: Clearly state that you collected data for 2014 or 2015 about proportion (percentage) of population 80+, social care expenditures for those 80+ (see my previous note about using one, consistent term if social care=home help), and median days in residential care for each municipality in the Dalarna region. Explain why 65+ for residential care versus 80+ for other variables. Para #4: I suggest: 'Based on 2014 (or 2015?) data from an annual national survey distributed to adults aged 65+ years receiving home help and residential care, we measured the quality of home help. We measured this variable as the percentage of respondents who answered positively to questions about two domains 1) response, trust, and safety of home help and 2) influence and adequate time.' Include information about how this survey was delivered, how data were collected, and what the response rate was. The descriptors need to be explained. Items used to collect data about "response" and "influence" would be helpful to include. Also, define what it meant to "answer positively." Note whether these data were aggregated based on municipalities or were individual-level data you could not link to participants. Note this limitation. Page 9: Para #1: I recommend removing "...as appropriate..." when referring to descriptive statistics. Explain in the section about data analysis or in the discussion about data collection, how you accounted for time as a factor in the analysis. Were any data censored? Page 10: Para #1: Present SD and range with mean for # chronic diseases (and other variables) and/or median if there were outliers. I cannot make sense of the range of home help receipt without a better understanding of how this variable was measured. Para #2: Remind the reader of the timing of data collection e.g., 'The mean number of primary care visits in 2014 or in the 12 months prior to an initial ED visit...' Para #3: Explain SEK and USD before using. Include the year in cost results. Use \$ before US dollar figures. Again, I cannot interpret a statement such as "Positive home help quality varied between 35.0 % and 56.0% for response, trust, and safety" without know more about this measure. Percentage of respondents endorsing items asking about...? Page 11: Para #2: Suggest removing "...the demographic and health characteristics..." Para #4: This statement is confusing "...while being admitted to hospital at initial ED visit was significantly negatively associated with an ED revisit." Wouldn't a 30-day revisit after initial ED visit be a problematic measure given the 30-day clock started after ED visit and not after hospital discharge? Or did you count 30-days from ED or hospital discharge? Page 11-12: In the presentation of findings for each model, it is difficult to easily read and consider the clinical significance of these findings with deviance statistics and pseudo-r-squares included. Can they be moved to table footnotes? I would like to see ORs with 95% CIs in the narrative rather than statements about "significant associations." I recommend at least one statement, in results and abstract, about the most clinically significant OR(s) e.g., 'There were greater odds of older adults having a 30-day revisit to the ED for those who received instrumental and personal care home help
--	--

	(OR=1.31, 95% CI=1.12-1.51) compared to those who received no home help.' Page 13: Para #1: Avoid new terms and citation format e.g., "Grunier et al. (2011) argued that proactive care can play a role in the management of need..." unless you are going to define and discuss the terms 'proactive care' and 'management of need.' I recommend you eliminate this sentence and reference. Para #2: Remove the first comma in the first sentence and revise "inadequate co-ordination between hospitals, primary and municipal social care..." Do you mean '...inadequate care coordination between hospital, primary care, and home help providers...'? Define integrated care. Para #3: "The ED visit model explains more variance in the outcome than the revisit model and more IVs are significant in the model, including contextual factors" does not make sense given the outcome was ED visits/revisits. Table #1 footnotes: I think you need to explain the analytic process you used to identify the protective factor of >80+ adults in the population as a suppressor variable for an ED visit elsewhere and provide a reference. This confused me. Thanks again for conducting this research!
--	--

VERSION 1 – AUTHOR RESPONSE

Reviewer: 1

Comments to the Author:

An interesting contribution to the existing literature on drivers for health service utilisation, in this case ED visits by older persons.

Response: Thank you for your positive valuation of our manuscript.

Reviewer: 2

Comments to the Author:

Overall, the authors address an important research topic: ED visits as an indicator of the quality of ambulant health care among the population in an age of 80+. To this end, they employ a quantitative model which explains ED visits by individual and, as a novel contribution from the perspective of the authors, contextual factors. The database refers to a region in Sweden which comprises municipalities with a high variance of demographic characteristics and other factors relevant for the model. The findings indicate that considering contextual factors improves the explanatory power of the model.

While the statistical analysis is carried out in a consistent way, some questions regarding the research setting and the interpretation of the results remain open and need to be addressed in further detail. First, while it is obvious that individual factors such as higher age and the number of chronic diseases are highly correlated with ED visits, the latter factor can also be an indication for poor quality of outpatient care. The direct and indirect mechanisms by which those factors might affect ED visits should be discussed therefore in detail when the results are presented.

Response: Thank you for this comment. We do agree with your suggestion and think that our use of term 'proactive' care to describe outpatient care lacked precision. Therefore, we have revised this text to provide greater clarity. In the revised manuscript, we have added the term 'multimorbidity' and replaced the term 'proactive care' with 'social and primary care' (Discussion section, second paragraph p.13).

Comment: Second, albeit the paper focuses on the relevance of context factors, only two of the six factors considered by the original model appear to be significant in the final model: distance of the place of residence to the ED and the proportion of 80+ residents in the population. That distance between home and nearest ED increases the odds of ED visit is an obvious and plausible result since

a smaller distance delivers a more convenient access. The interpretation of this findings seems to be however a bit vague and overstretched:

What consequences follow from the general statement that access to care should be driven by need rather than by place of residence? Shall we supply more or less EDs? What can we learn from the finding if distance is always a factor affecting the utilization of EDs even if distance is small?

Response: We have revised the discussion of contextual factors and now discuss distance to the ED and ED use in terms of appropriateness of such visits and accessibility of primary and social care (p 15).

Comment: That the proportion of 80+ residents is a predictor for a higher chance to ED visits is an interesting hint. One might suspect that a higher share of elderly in a municipality raises healthcare costs and, hence, reduces per capita expenses on health care. However, the annual budget costs for social care per person do not appear significant in the statistical analysis. The reader wonders therefore why the proportion of the 80+ residents in the population affects the odds of ED visits. The authors however do not offer an interpretation for their finding.

Response: Thanks for highlighting that there was no interpretation of this finding in the manuscript. We have included an interpretation of this finding in the revised version (p 14-15). Regarding costs, the statistical model included only the annual cost for social care and not for health care.

Comment: To strengthen the results, some recommendations are made:

Including Frequent User of ED (FUED) as indicator for a poor quality of outpatient care FUED (min. 5 visits per year) account for a disproportionately high number of all ED visits and are more likely to be associated with the lack of sufficient outpatient medical treatment, while one revisit might be random. Therefore, the proportion of FUED could enhance the interpretation of the results and should be also covered by the analysis as an additional outcome variable.

Response: This is an interesting observation. However, the mean number of ED revisits in our data was low, with low SDs, with ED revisits occurring for less than 1 in 5 patients across municipalities. FUED vs. non-FUED would therefore be a highly skewed DV given that the number of patients meeting the threshold of five visits would be very low. This might not create a problem for the individual-level variable estimates in the model given our sample size, but would be problematic for the contextual-level variable estimates given the amount of variance in ED revisits explained at that level was very low, and would be anticipated to be even lower in a DV with less variance to explain. Using FUED as an outcome variable would also mean retrospectively altering our research questions as explaining FUED was not the original purpose of our study.

Comment: Stratifying the age group of the 80+ population into further age brackets, e.g., age 80-84, age 85-89, age 90+. Note that the health situation might differ largely across these groups and, hence, the need for health care and ED visits. Considering further age brackets by the analysis could therefore deliver also interesting insights on the context factors, e.g., variables which appear insignificant for the aggregate 80+ population, might appear significant in subgroups. As an example, a community with a higher proportion of elderly residents might include more high-aged people who by nature suffer from poorer health which in turn might lead to a higher frequency of critical conditions.

Response: Age-stratified analysis is a good suggestion, but for a number of reasons we would not wish to pursue this option at present. First, data on contextual-level factors are available to us only at the age cut-offs indicated. Thus, in models for those aged e.g., 80-84, 84-85, 90+, whereas the data for the individual-level variables would be different across models as we could select individuals by age, the data for the contextual-level variables would be invariant across models. It is not clear therefore what anybained differences in the results across the age-stratified models would tell us about the relative value of contextual-level factors. Second, the age-stratified models would diminish in power due to the reduction in sample size from 80-84 through to 90+, and so variations in results across the models could be an artifact of, or at the very least influenced by, the reduction in power. Third, it is not clear on what basis the age-stratification should be made - e.g., what is the argument for combining an 85-year-old with an 89-year-old as opposed to with an 84-year-old? This

would need further thought and justification. We believe 80+ is a relevant cut-off of relevance to social care receipt and ED use in the Swedish context, but the issue you raise is interesting and we would hope to explore age-stratified analyses in future studies.

Reviewer: 3

Comments to the Author:

Thank you for this interesting paper. I did have a little difficulty with 'contextual factors', lying as they do at the heart of your theory. The Andersen paper is the final phase in developing models for studying community health, and the model (using aggregate data) "divides the major components of contextual characteristics in the same way as individual characteristics have traditionally been divided—those that predispose (e.g., community, age structure), enable (e.g., supply of medical personnel and facilities), or suggest need for individual use of health services (e.g., mortality, morbidity and disability rates). It would help the reader if the introduction explained this, and if you structure the introduction as you have done in the summary (the second paragraph of page 6). This would foreshorten and clarify this important section.

It also seems that primary and social care are central to the model.

Response: Thank you for this reflection. We have included a brief description of the Andersen model in the revised manuscript (see introduction para 2, p.4).

Comment: In the methods, you use the Cox and Snell and the Nagelkerke R^2 models to assess goodness of fit and improvement in the model, but how much of the variance do you actually explain?

Response: As explained in the manuscript (see p. 10), the R^2 statistics you describe approximate to the R^2 statistics for multiple regression. Thus, they identify the variance explained in the DV. As the two R^2 statistics differ in their methods of calculation, we provided both.

Comment: In the discussion, it would help to structure it as main findings, strengths and limitations (as you have) comparison with other studies and then meanings and or implications for practice. What are these?

Response: Suggested subheadings have been included in the revised manuscript, together with additional text on the implications for practice (p. 15).

Comment: It seems to this reviewer that the main finding was that instrumental services and personal services are associated with visits to the ED, but can you really say that this suggests unmet needs? I think not, and you can't really say that it indicates a poorly functioning health service.

Response: Perhaps the term 'proactive care' in the introductory lines of the discussion on individual level factors was ambiguous. We have replaced the term 'proactive care' with 'primary and social care' (p. 13). Research (Grunier et al., 2011; Or & Penneau 2018) has found that primary care and social care are important in managing the needs of older persons and in reducing ED visits, while the finding that the needs of frequent/heavy users of primary care are not met (Cunningham et al., 2017) suggests poor quality/functioning of primary care.

Comment: In strengths and limitations, you should mention the low proportion of the variance that you actually explained. Thank you again.

Response: Thanks you for your constructive comments. We have revised the text in line with your suggestion (p.16).

Reviewer: 4

Comments to the Author:

Thank you for conducting this interesting research. Below are my questions and suggestions with your statements in double- and my suggested revisions in single-quotation marks.

Comment: Abstract: Explain ED before use and/or use acronym only in text. Revise

Participants: 'Participants were 16,543 community-living adults aged 80 or older who were residents of the Dalarna region of Sweden, excluding older adults who moved out of Dalarna or into residential care during the study period.' Use the term participants, rather than sample, throughout.

Response: Suggested changes have been made in the revised version.

Comment: Include exposure/predictor variables as well as outcome variables in the abstract.

Response: Thank you for your suggestion, but we have followed the journal's guidelines on the structure and subheadings of the abstract (Objectives; design; setting; participants; interventions (if any); primary and secondary outcomes; results; conclusions).

Comment: Wasn't the main outcome of interest one or more ED revisits within 30 days of an initial ED visit?

Response: There were two outcome measures: (1) initial ED visit, and (2) ED revisit within 30 days of initial ED visit.

Comment: Results: 'Approximately 36% of participants visited the ED during the study period with 19% returning to the ED within 30 days.' Do not include p-values in the abstract; rather, present the effect size with 95% CI. 'In final models, an ED visit was associated with...while a return ED visit was...' Explain in the variables section of the abstract what you mean when you refer to "proportion of 80+ in the population" and "disposition at initial ED visit."

Response: Suggestions on language editing and explanation of variables have been considered and revised accordingly. Regarding use of p-values, we have presented p-values of model fit to present the contribution of contextual factors in explaining variance in the outcome. Therefore, we think that the presentation of p-values is relevant for the aim of this study.

Comment: 'Contextual factors in addition to individual factors improved prediction of return ED visits among older adults. These findings may indicate inequitable access to ED care and suggest the need for local support to improve health-related outcomes among older adults.'

Response: Thanks for this comment, but original text represents results, and is based on the interpretation of findings, for example, "Access to ED care should be driven by need rather than place of residence, and the variance in initial ED visit explained by contextual factors challenges the idea of universalism and equitable access to health care" (p. 15).

Comment: Strengths and Limitations: 'Contextual factors that explain ED use were included as predictors of returning to the ED within 30 days; this was a register-based cohort study, with robust information about the health status and health and social care use for the entire older adult population; receipt of home help was measured using both type and quality of home help care as predictors; administrative differences in how municipalities report data may impact data reliability.'

Response: Suggested changes have been made, except for the first bullet point which is not the case.

Comment: Keywords are not represented in the abstract. Community-living older adults? Contextual predictors of ED use? 30-day ED revisits?

Response: Such keywords are presented in the title, keywords should differ from those terms used in the title to increase the number of terms/phrases that might be used in literature searches.

Comment: Page 4: For the first reference, consider citing a more recent databased-paper rather than a 2012 review article about ED crowding (but not the appropriateness of use). I recommend 'Researchers conducting a systematic review found...' Similar to your first citation, the second is not based on recent data; I suggest this recent systematic review <https://bmccgeriatr.biomedcentral.com/articles/10.1186/s12877-019-1197-9>

Response: Thanks for the suggestion, but suggested reference is about the risk factors of frequent ED visits which is beyond the scope of this manuscript. Therefore, have not made the suggested change.

Comment: Remove the statement "which is often poorly adapted to such needs and may carry a risk of negative outcomes" if the meaning is explained in the sentences following. Otherwise, explain further the risk for negative outcomes. Note: ED providers offer services/treatments, not EDs—avoid other anthropomorphism e.g. 'We explored' rather than "this study explores." Based on the outcome of interest identified in the abstract, I expected a purpose statement such as 'We explored whether contextual in addition to individual factors explained a return visits to the ED within 30 days for adults ≥80 years of age.' I recommend removing the sentence about your study aim since it is described at the end of the introduction.

Response: Regarding the statement 'poorly adapted...outcomes', we think that this statement provides a background/explanation to the latter part, that is, concerning the risk of negative outcomes. Thank you for the recommendation on avoiding informality. A brief statement of the study's aim at an

early point of the Introduction facilitates readers' understanding of the paper's focus, and so we have retained that statement in the revised manuscript.

Comment: Please briefly describe "the well-established Andersen model" for those not familiar with the model. You may want to add a statement such as 'we used the Anderson model as the conceptual framework for this research.'

Response: A brief description of the Andersen model is included in the revised manuscript (see Introduction para 2 'Briefly, contextual factors.....disability rate'p.4).

Comment: Rather than "different care services," focus on your outcome of interest, 'ED use.' I do not think examples are necessary for the next sentence. Again, no need to discuss gender differences in primary care use, unless you link it to ED visits/revisits. Say more about the predictors of ED revisits, as described by de Gelder et al. (2018). Their findings are most relevant to your research. Though the data collected by Hass et al. (2015) are old, their study is also worthy of mention in this section. Their outcome of interest was similar—the number of ED admissions in a year, and their primary independent variable of interest, unmet ADL needs, it seems to me, could be defined as lack of receipt of home help.

Response: Thanks for your suggestions on articles to cite, but the suggested articles are already cited in the Introduction (References: 6; 8).

Comment: Perhaps 'logical' or 'commonsense' instead of "appropriate" when discussing the link between the need for (home help) care and (ED and hospital) services?

Instead of "Low staff competence and (lack of) continuity (of home help services)," given the information you offer next, I recommend 'A limited number of home help providers and poor continuity of care...' You may want to cite Bravell et al. (2021).

Response: Revisions have been carried out according to your suggestions except for rephrasing as 'A limited number' because the focus was competence and not number of providers (p.4,5). A reference to Bravell et al. (2021) has been added (p. 5).

Comment: Page 5: Para #1: You offer helpful, local information about available home health nurses in Sweden vs. Norway. However, the reference (based only on title) does not seem applicable ("International COVID-19 care in residential care.") Is the source not your reference #3?

Response: Cited references are correct. The reference 'International experiences of COVID-19' provides up-to-date comparison with neighbouring countries (e.g., social care for older adults, number of personnel, competence etc.) while discussing the pandemic. Reference 3 is a national status report of social care for older adults.

Comment: Do the terms "municipal social care" and "home help" mean the same thing? If they do, I would use home care or home help consistently.

Response: Social care includes residential care and home help which means that home help is a component of municipal social care. Therefore, we have used social care and home help as appropriate.

Comment: Readers may not know what a fortnight is. Instead of "However, studies on health care use have focused on home help receipt per se, rather than its level..." I recommend: 'Most researchers investigating the use of health care services among older adults measure home help receipt per se, rather than type and quality of home help.' You cite your previous work for this statement so perhaps 'We and other researchers investigating...'instead.

Response: We have revised the text accordingly (p.4).

Comment: Para #2: Add 'services' to the end of the first sentence. Instead of "...municipalities decide on eligibility criteria for services provided via home help and residential care," I recommend '...municipalities decide on eligibility criteria for home help and residential care services.' Revise as well '...driven by their need rather than their place of residence.' Based only on a review of the abstract, it does not appear that reference #19 supports your claim. Is there any more recent research about the variability of services for older adults in Sweden?

Response: We think the relevance of the article you mention is relevant. Please check the cited article, for example the first paragraph in the Discussion: 'We contrast our present findings to previous studies. These suggested that the extensive local variation in the service distribution made

the concept of diverse welfare municipalities more adequate in describing elder-care services in Sweden, and that the universalistic principle of equal access to services for all citizens across the nation was threatened by this local variation.'

Comment. Para #3: Remove the reference to "international changes" in the first sentence since your focus is on Europe. Revise the second sentence so it is two sentences or phrases joined with a semi-colon—the first about hospital use and the second about residential care use. Revise "...homehelp provision has not fully compensated for the decrease in residential care." I recommend 'Use of home help services has not increased relative to the decreased use of residential care among older adults.' Reference #18, though published in 2012, is based on data that are >15years old. Is there no more recent reference?

Response: Our focus is not only on Europe. We have used it as an example. Regarding rephrasing of home help provision, your suggested changes alter the meaning from provision to use, which would not be correct.

Comment. Para #4: I suggest the topic sentence should be 'It is important to explore contextual factors potentially influencing ED use among older adults.' Refer to prior study findings in past tense e.g. 'Researchers in France found...was associated with...' Use a similar format for other researchers' findings e.g., 'Researchers in England found limited social care services predicted increased ED revisits.'

Response: We have revised the text by providing the authors' names.

Comment. Page 6: See my prior comment about your purpose statement. I thought the outcome of interest was 30-day ED revisits? Perhaps add a secondary aim about identifying predictors of ED revisits.

Response: We have rephrased ED visits to 'initial ED visit' to make it clearer that there are two outcome variables.

Comment. Para #3: Revise "...resident in the Swedish..." to '...who were residents of the Swedish...' Specify that there were no exclusion criteria other than moving out of the area or into residential care. Specify how and how often participant follow-up data were collected. Include end date for the study period.

Response: Language editing has been carried out (p.6). Regarding the end of follow-up, the outcome variables are described so: 'ED visits during 2015 and ED revisits within 30-days of initial ED visit'. Since time is not included as an explanatory factor, exact dates would be an unnecessary detail for the reader.

Comment. Para #4: Instead of "All hospital EDs are 24-hour operational, one regional (Falun) and three local (Mora, Avesta, and Säter; Säter is for psychiatric care patients only)" I suggest 'There are 4 hospital EDs in Dalarna, all open 24-hours; one is regional, three are local, and one is only for psychiatric emergencies.' If these services are not long-standing or have changed, I recommend 'During the study period, there were 4...'

Response: Changes have been made according to your suggestions (p.7).

Comment. I do not understand the need for section 2.2. Use consistent upper- or lower-case titles for all headings/subheadings.

Response: This section is mandatory according to the author guidelines of BMJ Open.

Comment. Page 7: I would discuss variables of interest before describing data sources. Then identify how each variable, with the level of measurement already defined, was extracted from the data source.

Response: Unfortunately, we do not understand this comment and have therefore not made any changes. We believe the detail provided in the description of our variables is satisfactory.

Are there 3 or 4 registers? Is the Kolada database the same as the Dalarna regional database?

Response: No, Kolada is a national database, whereas the health care database of Dalarna is a regional register.

Comment: Address any bias in terms of data collection or measurement of variables of interest e.g. you mention in the abstract that municipal data may vary across regions. Explain further here.

Response: We apologise for being unclear in the abstract. The local variation refers to contextual factors, not the quality of variables. We have also addressed potential threats to reliability in the strengths and limitations sub-section of the Discussion.

Comment: Re DVs: Specify whether ED visits included visits to any ED e.g., psychiatric ED. Since you did not measure the distance to psychiatric ED, I assume you did not include psychiatric ED visits. Explain.

Response: We did include visits to psychiatric ED, but given the specialisation of the psychiatric ED, distance to this ED is not relevant for visits other than those concerning psychiatric emergencies. To clarify that all EDs were included, the text has now been revised as 'Information on visits at four EDs was obtained from the regional database' (p.7).

Comment: Re IVs: Include measurement of each demographic variable e.g., age in 2014 based on date of birth. Revise 'We gathered information about the health of participants by collecting ICD 10 diagnoses for all inpatient and outpatient visits in 2014 from the regional database. We measured the health of participants as the number of ICD codes for chronic diseases.' Note this limitation (that data were based only on 2014 healthcare utilization) elsewhere.

Response: Suggested changes are being made to the age variable (p.8) and a reflection added to the strengths and limitation sub-section of the Discussion (p.16).

Comment: Page 8: Specify how often data were collected for receipt of home help during 2014 or 2015. Were data collected about # of home help visits? Type of home help provider? Over what period of time? Until now, I assumed home help included nurse home visits. If this is not the case, it would be helpful to define home help as excluding nurse home visits earlier in the paper.

Response: Home help status is measured as approval/decision by the local authority on home help provision. This information is included in the revised manuscript p 8. Home help is briefly defined in the Introduction '(instrumental services, personal care or both) p 5.

Comment: Para #2: I don't understand the statement "...the latter IV only analysed in relation to the ED revisit DV." Data about disposition at initial ED visit was collected only for participants who had a 30-day ED visit?

Response: Yes, the disposition variable was an IV only in the model on ED revisit.

Comment: Para #3: Clearly state that you collected data for 2014 or 2015 about proportion (percentage) of population 80+, social care expenditures for those 80+ (see my previous note about using one, consistent term if social care=home help), and median days in residential care for each municipality in the Dalarna region. Explain why 65+ for residential care versus 80+ for other variables.

Response: The proportion of population 80+ was measured at 2015 and we have included this information on p 8. Home help is a part of social care, which also covers, e.g., care homes. The reason for using 65+ rather than 80+ for median days at residential care was due to the availability of data.

Comment: Para #4: I suggest: 'Based on 2014 (or 2015?) data from an annual national survey distributed to adults aged 65+ years receiving home help and residential care, we measured the quality of home help. We measured this variable as the percentage of respondents who answered positively to questions about two domains 1) response, trust, and safety of home help and 2) influence and adequate time.' Include information about how this survey was delivered, how data were collected, and what the response rate was. The descriptors need to be explained. Items used to collect data about "response" and "influence" would be helpful to include. Also, define what it meant to "answer positively." Note whether these data were aggregated based on municipalities or were individual-level data you could not link to participants. Note this limitation.

Response: Thank for these suggestions. We have included the questions asked in the revised manuscript (p. 9) and we have provided references for further details.

Comment: Page 9: Para #1: I recommend removing "...as appropriate..." when referring to descriptive statistics. Explain in the section about data analysis or in the discussion about data collection, how you accounted for time as a factor in the analysis. Were any data censored?

Response: We have removed 'as appropriate' in the revised manuscript. Regarding time, it was not considered as a continuous variable in our analyses, rather our ED visit/revisit outcomes were binary (yes/no). ED visit had to occur within 30 days of initial ED visit. This information is provided in the manuscript.

Comment: Page 10: Para #1: Present SD and range with mean for # chronic diseases (and other variables) and/or median if there were outliers. I cannot make sense of the range of home help receipt without a better understanding of how this variable was measured.

Response: In the text, mean is presented for the total sample along with ranges across municipalities. Tables present mean and SD stratified by municipalities. Home help is used as a categorical variable as described in the Method section.

Comment: Para #2: Remind the reader of the timing of data collection e.g., 'The mean number of primary care visits in 2014 or in the 12 months prior to an initial ED visit...'

Response: This information is provided in the Method section and in our view repeating the information would inflate the word count unnecessarily.

Comment: Para #3: Explain SEK and USD before using. Include the year in cost results.

Use \$ before US dollar figures. Again, I cannot interpret a statement such as "Positive home help quality varied between 35.0 % and 56.0% for response, trust, and safety" without know more about this measure. Percentage of respondents endorsing items asking about...?

Response: We have revised the manuscript according to your suggestions (see p.10). Regarding positive home help quality see our response to the comment above on measuring of positive home help.

Comment: Page 11: Para #2: Suggest removing "...the demographic and health characteristics..."

Response: We believe that these words help the readers to understand what the model includes and have not made the suggested change.

Comment: Para #4: This statement is confusing "...while being admitted to hospital at initial ED visit was significantly negatively associated with an ED revisit." Wouldn't a 30-day revisit after initial ED visit be a problematic measure given the 30-day clock started after ED visit and not after hospital discharge? Or did you count 30-days from ED or hospital discharge?

Response: Measuring ED revisit within 30-days of initial ED visit is in line with previous research (Lowthian et al., 2016; McCusker et al., 2000). We count 30days from ED visit. We do agree with your concern, and we have reflected on this in our strengths and limitations sub-section of the Discussion.

Comment: Page 11-12: In the presentation of findings for each model, it is difficult to easily read and consider the clinical significance of these findings with deviance statistics and pseudo-r-squares included. Can they be moved to table footnotes? I would like to see ORs with 95% CIs in the narrative rather than statements about "significant associations." I recommend at least one statement, in results and abstract, about the most clinically significant OR(s) e.g., 'There were greater odds of older adults having a 30-day revisit to the ED for those who received instrumental and personal care home help (OR=1.31, 95% CI=1.12-1.51) compared to those who received no home help.'

Response: Thank you for this comment, but we think that presenting model fitting statistics is relevant with regards to the aim of the study. We also seek to be consistent in the presentation of results, OR or model statistics, rather than presenting OR only for one or two variables.

Comment: Page 13: Para #1: Avoid new terms and citation format e.g., "Grunier et al. (2011) argued that proactive care can play a role in the management of need..." unless you are going to define and discuss the terms 'proactive care' and 'management of need.' I recommend you eliminate this sentence and reference.

Response: The term 'proactive care' has been removed from the revised manuscript.

Comment: Para #2: Remove the first comma in the first sentence and revise "inadequate co-ordination between hospitals, primary and municipal social care..." Do you mean '...inadequate care coordination between hospital, primary care, and home help providers...'? Define integrated care.

Response: Your suggested changes have been made. Integrated care is co-ordination between different care providers. We have included a brief definition (p.14).

Comment. Para #3: “The ED visit model explains more variance in the outcome than the revisit model and more IVs are significant in the model, including contextual factors” does not make sense given the outcome was ED visits/revisits.

Response: As mentioned earlier, there are two models with different outcome variables. This statement is comparing the two models.

Comment. Table #1 footnotes: I think you need to explain the analytic process you used to identify the protective factor of >80+ adults in the population as a suppressor variable for an ED visit elsewhere and provide a reference. This confused me.

Thanks again for conducting this research!

Response: We used the standard method for identifying a suppressor variable (for details, please see Field, A. Discovering statistics using IBM SPSS Statistics. 4th Ed. London: SAGE Publications, 2013). The book is cited in the footnote of Table 4.

VERSION 2 – REVIEW

REVIEWER	Schmiedhofer, Martina Charite Universitatsmedizin Berlin, Arbeitsbereich Notfall- und Akutmedizin
REVIEW RETURNED	05-Dec-2021

GENERAL COMMENTS	Thank you for thoroughly revising your manuscript. All my recommendations are either considered in the revision or sufficiently adressed.
---

REVIEWER	McBride, David University of Otago, Department of Preventive and Social Medicine
REVIEW RETURNED	06-Dec-2021

GENERAL COMMENTS	Thank you for this careful revision and your responses.
---